# Bioavailability of Lutein from Marigold Flowers (Free vs. Ester Forms): A Randomised Cross-Over Study to Assess Serum Response and Visual Contrast Threshold in Adults

**DOI:** 10.3390/nu16101415

**Published:** 2024-05-08

**Authors:** Begoña Olmedilla-Alonso, Fernando Granado-Lorencio, Julio Castro-Feito, Carmen Herrero-Barbudo, Inmaculada Blanco-Navarro, Rocío Estévez-Santiago

**Affiliations:** 1Instituto de Ciencia y Tecnología de Alimentos y Nutrición (ICTAN-CSIC), c/José Antonio Novais, 6, 28040 Madrid, Spain; 2Hospital Universitario Puerta de Hierro-Majadahonda, c/Maestro Rodrigo, 2, 28222 Majadahonda, Spain; granadof@hotmail.es (F.G.-L.); maria.herrero@cfppuertadehierro.org (C.H.-B.); macublanco10@gmail.com (I.B.-N.); 3Optometric Center Zires, c/Juan de Austria, 16, 28016 Madrid, Spain; jcastro@zires.es; 4Facultad de Ciencias de la Salud, Universidad Francisco de Vitoria, Ctra. Pozuelo-Majadahonda Km 1800, 28223 Pozuelo de Alarcón, Spain; rocio.estevez@ufv.es

**Keywords:** lutein intake, blood lutein, lutein free, lutein ester, contrast sensitivity, zeaxanthin

## Abstract

Lutein (Lut) and zeaxanthin (Zeax) are found in the blood and are deposited in the retina (macular pigment). Both are found in the diet in free form and esterified with fatty acids. A high intake and/or status is associated with a lower risk of chronic diseases, especially eye diseases. There is a large global demand for Lut in the dietary supplement market, with marigold flowers being the main source, mainly as lutein esters. As the bioavailability of Lut from free or ester forms is controversial, our aim was to assess the bioavailability of Lut (free vs. ester) and visual contrast threshold (CT). Twenty-four healthy subjects (twelve women, twelve men), aged 20–35 and 50–65 years, were enrolled in a cross-sectional study to consume 6 mg lutein/day from marigold extract (free vs. ester) for two months. Blood samples were taken at baseline and after 15, 40, and 60 days in each period. Serum Lut and Zeax were analysed using HPLC, and dietary intake was determined with a 7-day food record at the beginning of each period. CT, with and without glare, was at 0 and 60 days at three levels of visual angle. Lut + Zeax intake at baseline was 1.9 mg/day, and serum lutein was 0.36 µmol/L. Serum lutein increased 2.4-fold on day 15 (up to 0.81 and 0.90 µmol/L with free and ester lutein, respectively) and was maintained until the end of the study. Serum Zeax increased 1.7-fold. There were no differences in serum Lut responses to free or ester lutein at any time point. CT responses to lutein supplementation (free vs. ester) were not different at any time point. CT correlated with Lut under glare conditions, and better correlations were obtained at low frequencies in the whole group due to the older group. The highest correlations occurred between CT at high frequency and with glare with serum Lut and Lut + Zeax. Only in the older group were inverse correlations found at baseline at a high frequency with L + Z and with Lut/cholesterol and at a low frequency with Lut/cholesterol. In conclusion, daily supplementation with Lut for 15 days significantly increases serum Lut in normolipemic adults to levels associated with a reduced risk of age-related eye disease regardless of the chemical form of lutein supplied. Longer supplementation, up to two months, does not significantly alter the concentration achieved but may contribute to an increase in macular pigment (a long-term marker of lutein status) and thus improve the effect on visual outcomes.

## 1. Introduction

Lutein and its structural isomer zeaxanthin are obtained exclusively from food. These carotenoids can be absorbed intact or undergo oxidative cleavage prior to absorption from the intestinal lumen and are transported in the blood by lipoproteins, associated with LDL and HDL lipoproteins. Lutein and zeaxanthin are preferentially deposited in the retina where, together with meso-zeaxanthin (thought to be formed from lutein in the retina), they form the macular pigment (MP). This specific intraocular deposition suggests a biological process governing their capture, deposition, and stabilisation in the macula, thought to be mediated by binding proteins [1].

Lutein and zeaxanthin are oxygenated derivatives of carotenes (xanthophylls), which are mainly provided by green plant foods, olive oil, eggs, and egg by-products [1]. In most fruits and some vegetables, lutein and zeaxanthin are found in free form and esterified with fatty acids [2,3,4]. Although xanthophyll esters are the main form in most fruits and flowers, there is much less information on the occurrence of carotenoid esters in foods of plant and animal origin compared to the large amount of data available on the free forms of carotenoids [5]. Moreover, in the diet, the main xanthophylls (lutein, zeaxanthin, and β-cryptoxanthin) are mainly obtained as ester forms, as described in Spanish and Brazilian populations, where they represent about 70% of the total dietary xanthophyll intake [5].

On the other hand, high dietary lutein intake and/or blood concentration is associated with a lower risk of several chronic diseases (development or progression), most notably age-related macular degeneration (ARMD) [6,7,8], as well as health benefits, such as improved visual and cognitive function [7,9,10,11,12,13]. A reduction in the risk of progression from intermediate to advanced ARMD with lutein-rich supplementation [8] and improvements in visual function have also been described [14,15]. In the eye, the MP filters (or absorbs) blue light, physically protecting the underlying photoreceptor cell layer from light damage, but also acts as an antioxidant to protect against the formation of reactive oxygen species [1,15].

Today, lutein is one of the most sought-after bioactive compounds in the global food supplement market, mainly due to its protective role in the retina, in relation to the cosmetic industry (e.g., sun light protection), and as a colour additive in food and feed [16,17]. The main commercial source of lutein for dietary supplements comes from marigold flowers (*Tagetes erecta*), which are rich in carotenoids, especially lutein, which is mainly present as lutein ester (90–99%) [4]. Although the esterification of xanthophylls seems to facilitate their bioaccessibility and the ester forms are more stable during food processing [4,18], the bioavailability of lutein from free or ester forms seems to be controversial. The final product in the blood is free lutein, as lutein esters are hydrolysed in the lumen of the small intestine before being taken up by the enterocyte [1]. However, lutein esters have also been described in the serum of normolipemic individuals who take lutein supplements and achieve serum lutein concentrations above 1.06 µmol/L (60 µg/dL) [19].

Although in general the bioavailability of xanthophyll esters seems to be equal or higher than their free forms [5,18], with regard to lutein, while some studies show that the lutein ester is better absorbed [20,21], others show the opposite, the free form being better absorbed than the ester form [22], or even no difference [23,24]. Therefore, as more studies on lutein bioavailability (free versus ester) are needed, preferably in subjects with well-defined characteristics and in studies acting as their own control, our aims were to assess lutein bioavailability, free versus ester forms, taking into account two potential confounding factors, sex and age, and to provide an approach to the assessment of visual function in the context of lutein supplementation.

## 2. Materials and Methods

### 2.1. Subjects and Study Design

A total of 24 apparently healthy subjects (12 women, 12 men) in two different age groups (20–35 and 50–65 years) were recruited according to the inclusion and exclusion criteria. Inclusion criteria were as follows: serum cholesterol ≤ 5.7 mmol/L, biochemical and haematological profile within normal limits, and no medication, vitamin, or dietary supplements. Exclusion criteria were as follows: habitual use of lutein supplements and cataract or any ocular disease affecting the macula. Volunteers were enrolled in a randomised cross-over study to take a lutein supplement (free or ester form) of 6 mg lutein/day for two months, followed by a two-month wash-out period (Figure 1). Participants were asked to continue their usual diet and to take lutein capsules with one of their daily meals.

Blood samples (8–10 mL) were collected after an 8–10 h fast at baseline and at days 15, 40, and 60 of each of the two intervention periods. Subjects underwent a 7-day food record at the start of each supplementation period. The visual system was assessed using contrast threshold (CT) at 0 and 60 days. This study was approved by the Clinical Research Ethics Committee of the Hospital Universitario Puerta de Hierro of Madrid, Spain (Dated 27 April 2007, code: CEIm.), and the subjects gave their written informed consent.

### 2.2. Lutein Supplements

Lutein capsules (6 mg/capsule) were manufactured by Goerlich Pharma International, GmbH (Edling, Germany) using Flora GLO 10% VG TabGrade^TM^ (Kemin, Des Moines, IA, USA) and from Xangold 10% (Cognis, Monheim am Rhein, Germany) for the capsules containing the free and ester forms of lutein, respectively. Both Flora GLO and Xangold are made from marigold flowers and contain low levels of zeaxanthin (a minimum of 0.4% in Flora GLO and approximately 4.9% in Xangold, according to the manufacturers). The overall composition of the free lutein capsules was as follows: Flora GLO 10% VG 22.22%, cellulose microcrystalline 77.04%, and magnesium stearate 2%, and that of the lutein ester capsules was: Xangold 10% 43.33%, cellulose microcrystalline 55.93%, and magnesium stearate 0.74%. The capsules were provided at each visit, with a few extra capsules each time.

### 2.3. Assessment of Dietary Intake of Lutein and Carotenoids

Dietary intake of carotenoids was assessed using 7-day food records at the beginning of each of the two intervention periods. Food records were kept for seven consecutive days (including one weekend) and were collected by a specialised dietician who reviewed the records in the presence of the participant, asking questions about quantities or items to facilitate correct identification of the food/drink consumed. Amount consumed was estimated in portions or household servings and in units (fruit) [25]. On the basis of this information, we calculated the dietary intake (grams/day), which was used to determine the daily intake of lutein and zeaxanthin using a database developed by our group and included in a software application for the assessment of individual dietary intake of carotenoids [26]. This database contains HPLC-generated data on the main dietary carotenoids, mainly from fruits and vegetables [27,28,29].

### 2.4. Analysis of Serum Concentrations of Lutein and Zeaxanthin

Serum concentrations of lutein and zeaxanthin were determined using HPLC using a system consisting of a model 6000 pump, a Rheodyne injector and a 2998 photodiode array detector (Waters, Milford, MA, USA), and a Spheri-5 ODS 5 μm column (220 mm × 4.6 mm (Brownlee Labs, Applied Biosystem, Santa Clara, CA, USA) with a guard column (Aquapore ODS type RP-18, PerkinElmer Inc., Drachten, The Netherlands). The mobile phases were acetonitrile–methanol (85:15; *v*/*v*), and acetonitrile–dichloromethane–methanol (70:20:10; *v*/*v*/*v*) in a linear gradient from 5 min to 20 min. Both mobile phases were stabilised with ammonium acetate (0.025 mol L^−1^) added to the methanol. The flow rate was 1.8 mL min^−1^, and detection was at a wavelength of 450 nm. All chromatograms were processed using Empower 2 software (Waters, Milford, MA, USA).

Blood samples were collected in plain blood collection tubes without anticoagulant. The serum was separated using centrifugation (630× *g*, 10 min). Baseline and supplemented serum samples from each subject were stored at 70 °C, processed, and analysed simultaneously to reduce analytical variability. Subjects were analysed in random order within seven months of collection, and one in six serum samples were analysed in duplicate. Carotenoid extraction was performed on fasting serum samples using a slight modification of a previously published method [30]. Briefly, 200 μL of serum was added to 200 μL of ethanol, vortexed, and extracted twice with 400 μL of hexane: dichloromethane (5:1) stabilised with 0.1 g/L butylated hydroxyltoluene. The organic phases were pooled, evaporated to dryness under a nitrogen atmosphere, reconstituted with 200 μL of a solution of tetrahydrofuran: ethanol (1:2), and injected (5 μL) into the HPLC system. Methanol, ethanol, acetonitrile, dichloromethane, ammonium acetate, butylated hydroxytoluene (BHT), and tetrahydrofuran were supplied by Panreac (Barcelona, Spain). Lutein (xanthophyll from marigold) was purchased from Sigma Chemical Co. (St. Louis, MO, USA), and zeaxanthin was purchased from Fluka Analytica (Sigma Aldrich, St. Louis, MI, USA).

Blood total cholesterol was analysed with colourimetric enzymatic assay (Cobas Integra 400 plus, Roche, Basel, Switzerland).

### 2.5. Visual Contrast Threshold

Visual function was assessed by measuring contrast threshold (CT) with and without glare (Contrast Glaretester, CGT-1000, Takagi Sciko Co. Ltd., Nagano, Japan) at three times during the study period (0, 40, 60 days). The CGT-1000 determined CT with and without glare using an automated strategy set for 6 sizes of annular stimuli with diameters ranging from 6.3° to 0.7° of visual angle. The 12 levels of CT range from 0.01 to 0.45. The luminance of the background on which the stimuli were presented was 10 cd/m^2^. Stimulus presentation lasted 0.2 s with 0.2 s intervals; test luminance had a glare of 40,000 cd/m^2^.

The lower the CT, the higher the contrast sensitivity level at which a subject was able to detect each spatial frequency. The reciprocal of the CT is called contrast sensitivity. Each subject was tested monocularly for CT, once with each eye and with spectacle correction if necessary. The test results were automatically printed on a single graph showing the sensitivity functions. CT results were reported as the mean of three levels of frequency data—low, medium, high—corresponding to the mean of 6.3° and 4° of visual angle (lower frequencies), 2.5° and 1.6° (medium frequencies), and 1.0° and 0.7° (high frequencies).

### 2.6. Statistical Analysis

Data are expressed as the mean and standard error, 95% confidence interval. The normal distribution of the data was assessed (Kolmogorov–Smirnov test). Serum lutein did not follow a normal distribution at any time point in the lutein-free group and at baseline in the ester group. CT and lutein data from both groups showed a normal distribution at baseline but not at 60 days, so Pearson and Spearman correlations were used. Non-parametric tests (Friedman) were used to compare the concentrations of the variables analysed in the groups (four groups according to sex and age), the Willcoxon test to compare the concentrations at baseline, and the Kruskall–Wallis test to compare the responses to supplementation. Although there were differences for sex and age in each of the four groups, no interactions were observed for these variables in the lutein chemical formula groups. A linear mixed model with repeated measures for time and lutein chemical formula was used with lutein and lutein/cholesterol as dependent variables and age, sex, time, and chemical formula as fixed factors. Pairwise interactions of all factors were analysed.

All reported *p*-values are based on a two-tailed test and compared at a 5% significance level. IBM SPSS Statistics, v.27 software was used for all statistical calculations.

## 3. Results

The subjects (*n* = 24, divided into two age groups: *n* = 12 are 20–35 years old (mean and SD: 25 ± 3 y) and *n*= 12 are 50–65 years old (mean and SD: 53 ± 4 y)) supplemented their usual daily diet with 6 mg of lutein (free and ester), which was three times their mean dietary intake of lutein + zeaxanthin at a baseline of 1.9 mg/day (±0.23 SE) (median = 1.4 mg/day). Compliance was assessed by counting the number of capsules returned at each visit, and it was greater than 85%. As the lutein capsules supplied also contained zeaxanthin, albeit at a very low concentration compared to lutein, serum zeaxanthin was also analysed. Table 1 shows the serum concentrations of lutein, lutein/cholesterol, and zeaxanthin at baseline and at each time point analysed during the study (*n* = 48). There were no differences in the concentrations of lutein, zeaxanthin, and cholesterol (mean and SD: 4.69 ± 0.90 and 4.71 ± 0.89 mmol/L) at the beginning of each of the supplementation periods.

The serum lutein concentration increased on day 15, reaching levels of 0.81 and 0.90 µmol/L for free and ester lutein, respectively. These increases, which averaged 2.4 times, were maintained throughout the intervention study (days 40 and 60) in each group. The serum zeaxanthin concentration also increased from 0.10 to 0.16 µmol/L, an average 1.7-fold increase (Table 1). Zeaxanthin increased on day 15 (*p* < 0.001) and continued to increase on day 40 (*p* < 0.034), and it was maintained until the end of the study (60 d). At each time point, there were no differences between the responses to lutein and zeaxanthin supplementation with the two chemical formulae.

In the whole sample, serum lutein concentrations differed between sexes (0.44 ± 0.24 and 0.31 ± 0.12 µmol/L, men and women, respectively, *p* = 0.026) and age groups (0.27 ± 0.07 and 0.48 ± 0.23 µmol/L, younger and older, respectively, *p* = 0.011) at the beginning of the two supplementation periods and on day 15 but not after 40 and 60 days of supplementation. Age and sex had a weak effect on the serum lutein response to the supplementation (free or ester lutein), with higher responses in men than in women and in the older group (50–65 years), although this did not reach statistical significance. Figure 2 shows the time course of serum lutein concentration using the lutein chemical formula for groups according to sex and age.

The CT data (mean ± SD, [median]), with and without glare, for the six different degrees of visual angle, grouped into three levels according to the frequencies low (6.3° and 4°), medium (2.5° and 1.6°), and high (1.0° and 0.7°), are shown in Table 2. In the total group and also grouped by age, CT showed no significant differences at baseline in any of the periods with free lutein and lutein ester, and no differences were found in the responses to lutein supplementation with the two chemical formulae at any time. CT showed statistical differences according to light level, being higher in the presence of medium- and high-frequency glare at both baseline and 60 days and also in the presence of low-frequency glare at 60 days (Table 2 and Figure 3). Although CT differed by age at medium and high frequencies, there were no differences after adjustment for age (20–25 vs. 50–65 years). CT showed no differences after 60 days of lutein supplementation except in the glare condition for the low frequencies (*p* = 0.008).

After 60 d of lutein supplementation, age-adjusted CT was different in the glare condition for the low frequencies (*p* = 0.008).

Dietary lutein plus zeaxanthin intake showed no significant correlation with serum lutein, lutein + zeaxanthin, and lutein/cholesterol concentrations or with the CT at baseline. For serum concentrations of lutein plus zeaxanthin, lutein, and lutein/cholesterol, their correlations with CT, without and with glare, at baseline and after 60 days of supplementation in the total group (both lutein groups, free and ester) are shown in Table 3. In the total group, statistically significant correlations were found at baseline between serum lutein or lutein plus zeaxanthin concentrations and CT at each of the three frequency levels, with and without glare, but no significant correlations were found between CT and lutein/cholesterol concentrations. After 60 days of lutein supplementation, in the total group, with and without glare, all correlations of CT with lutein plus zeaxanthin and also with lutein were maintained except those under glare at low frequency and without glare at high frequency. The highest correlations were observed with glare and at the highest frequencies (lower visual angle of the estimulus) with serum lutein (0.322, *p* = 0.001) and lutein plus zeaxanthin (0.340, *p* < 0.001).

Comparing the age groups, at baseline in only the older group and in the glare condition, inverse correlations were found at high frequencies with lutein plus zeaxanthin (−0.343, *p* = 0.017) and with lutein/cholesterol (−0.390, *p* = 0.006) and at low frequencies with lutein/cholesterol (−0.326, *p* = 0.024). Instead, after 60 days of lutein supplementation, correlations were found only in the younger group (20–30 years), while no significant correlations were found in the older group. In the younger group, significant correlations were obtained without glare at low and medium frequencies with serum lutein (0.308, *p* = 0.035; 0.315, *p* = 0.021, respectively) and with glare at high frequency with serum lutein (0.367, *p* = 0.010) and with serum lutein/cholesterol (0.423, *p* = 0.003). Appendix A shows the correlations of serum lutein with CT, with and without glare, according to the age of the subjects at baseline and at the end of the lutein supplementation period, and an improvement can be observed at high and medium frequencies, mainly with glare.

## 4. Discussion

### 4.1. Serum Lutein and Zeaxanthin and Effect of Lutein Supplementation (Free vs. Ester)

This study of the bioavailability of lutein from dietary supplements focused on the forms of lutein (free vs. ester) found in commercially available supplements. Both forms are derived from marigold flowers and therefore also contain its structural isomer, zeaxanthin (ester forms), although at much lower concentrations [31]. The concentration in the supplements was set at 6 mg lutein/day because this amount has been associated with a reduced risk of several chronic diseases [6,32,33,34], although there is evidence that lower levels of lutein and zeaxanthin may be sufficient to protect against the progression of age-related macular disease [35]. The amount of 6 mg lutein can be obtained from the diet [33] by choosing lutein-rich foods (e.g., about 100 g of spinach), but it can be difficult to maintain on a regular basis, and its bioavailability from fruits and vegetables is low and generally lower than that from supplements [17].

The 6 mg lutein/day provided is much higher than the mean dietary intake of lutein + zeaxanthin consumed by the volunteers in this study (1.9 mg/day), which is similar to data from different adult populations/groups from Europe, the USA, and other countries worldwide, with ranges from 0.1 to 4.8 mg/day (mean of 2.2 mg/day), with 1.7 mg/day being the mean of the data from Spain [36]. These dietary intakes of lutein plus zeaxanthin corresponded to a blood concentration range of 0.20–0.56 µmol/L (mean 0.33 µmol/L) [36], similar to the range (0.43–0.58 µmol/L) but lower than the mean (0.51 µmol/L) found at the baseline in our study. Dietary and blood lutein concentrations are higher than those of zeaxanthin, being about four times higher in the blood in this study, which is consistent with the 4 to 5 ratio described in people from different countries [36]. In the Spanish diet, lutein is about 11 to 13 times higher than zeaxanthin [37,38].

The daily capsules contained a marigold flower extract rich in lutein (free or ester forms) and with a much lower concentration of zeaxanthin (companies reported from 0.4 to around 4.9%). Their consumption led to an increase in both xanthophylls in the serum, reaching a plateau after 15 days for lutein and 40 days for zeaxanthin. A similar plateau has been reported to be dose-dependent in some studies [39] but not in others [40,41]. Our data on serum lutein responses (6 mg/d resulted in an increase of 2.4) are comparable to those found in similar studies in healthy subjects, e.g., supplementation with 15 mg/d (extracted from marigold) increased serum lutein 5-fold and zeaxanthin approximately 2-fold [42] and supplementation with 5 mg lutein (ester from Tagetes erecta) increased serum lutein 2.6-fold [39]; however, this differs from those reported by Machida et al. [41], who found that a daily intake of 12 mg lutein (free form) for 16 weeks provoked a maximum two-fold increase in serum.

Lutein supplementation increased serum lutein concentrations to around 0.88 µmol/L, which is above the 0.60 µmol/L concentration associated with a lower risk of chronic eye disease in epidemiological studies [32,33]. In healthy individuals, this serum lutein response is influenced by inter-individual variability and the large variability in bioavailability due to the carotenoid source and host factors [36]. Among the potential confounding factors, blood lipid concentrations are not always considered inclusion or exclusion criteria, although lutein and zeaxanthin are transported by lipids and could therefore condition the response [37]. Other confounding factors, such as sex and/or age, have been described in observational studies [36,37], and the literature generally shows results adjusted for sex and/or age. However, some studies also reported no influence of these factors on the blood lutein response [20,22,39,42], which is consistent with our data, as the serum lutein concentration differed between sexes and age groups at baseline and in the serum lutein response until the plateau was reached, but their effect did not reach statistical significance, even when the increase was expressed as lutein/cholesterol.

Lutein and zeaxanthin are ingested in the diet in free or in ester form. While they are found in free form in green leaves (vegetables), they are esterified with many fatty acids in some leaves and tubers and especially in many ripe fruits [1]. In general, the bioavailability of the ester forms seems to be higher than that of the free forms due to a better solubilisation and extraction during digestion [4,18]. Therefore, a higher bioavailability could be expected from the intake of lutein supplements as the ester form compared to the free form. However, information on the carotenoid ester content in foods and food supplements is scarce [1,5]. On the other hand, commercially available lutein-rich food supplements contain lutein in both forms, free or ester, but few studies have evaluated their bioavailability in healthy subjects, and moreover, they have shown discrepant results. A higher response to lutein ester vs. free supplementation was described in some of the studies [20,21,24], but in only one of them, the difference was statistically significant [21]. In our study, a higher serum lutein concentration was also obtained in response to lutein ester supplementation but not significantly different from that obtained with free lutein. The opposite, a higher response to the free lutein supplementation, has only been reported in one study [22]. These studies differed mainly in the concentration of lutein administered (10, 12.5, and 20 mg/day), design (randomised, parallel) [20], study duration (single dose, 1 to 6 months), and sample size (most of them around 20 and the one, which showed a higher response to free lutein, with 72 subjects). In the present study, with a crossover design to minimise inter-individual variability in responses and with a lower amount of lutein supplied (closer to that present in food intake), the serum lutein response was proportional to those from the above-mentioned studies. In this study, serum lutein responses are independent of age and sex and almost independent of serum lipids, as described in other studies [20,22].

With regard to zeaxanthin, which was also provided in this study at much lower concentrations than lutein, a higher blood response to esterified than to free zeaxanthin has been described [43].

On the other hand, the dissolution of the lutein formulation has an important influence on its bioavailability [21]. For example, Bowen et al. [20] used a single dose of lutein (free and ester forms with different formulations: crystalline vs. powder) and concluded that the lutein diester formulation was more bioavailable than the free formulation but that the difference in formulation could have led to different dissolution in emulsion lipid droplets and ultimately different bioavailability.

### 4.2. CT and the Effect of Lutein Supplementation

Lutein and zeaxanthin are transported in the blood by lipoproteins and deposited in the retina, where they form, together with mesozeaxanthin, the macular pigment (MP), which acts as a blue light filter and can be considered a marker of long-term dietary exposure. The MP increases with lutein and zeaxanthin supplementation and is associated with improvements in contrast sensitivity and visual performance [15,40,44,45,46,47,48] in a dose-dependent manner (at each of the spatial frequencies of 3, 6, and 12 cycles/degrees) in healthy subjects [46] and in subjects with impaired visual function [9]. In this study and mainly under glare conditions, in the whole sample and grouped by age, as the serum lutein concentration increases, the CT value decreases, especially in the medium and high frequencies (Figure 3), and therefore, the contrast sensitivity increases.

CT was influenced by lutein supplementation, mainly under glare conditions and at low frequencies, and as in previous studies, this relationship appears to be age-dependent [47], found only in the older group, where an increase in the CT (lower contrast sensitivity) was obtained after two months of lutein supplementation. However, in the study by Yao et al. [49], a slight trend towards an improvement in the glare sensitivity (mainly at medium frequencies) was observed from the first to the third month of supplementation (20 mg/day), and only after six months of lutein supplementation was a significant change observed in healthy subjects. Also, in young subjects without chronic eye disease, Ma et al. [50] reported a trend toward higher thresholds at low and medium frequencies after three months of supplementation with 6 mg lutein/day but without improvement in glare conditions. Another study in healthy subjects (20–69 years) supplemented with 12 mg lutein/day for four months showed an improvement in contrast sensitivity (low frequency) and glare sensitivity (high and medium frequency) [41]. In subjects with age-related macular degeneration, lutein supplementation at different doses (6, 10, and 20 mg/d) with visual function assessments at six, nine, and twelve months showed no significant changes until the twelve months [15,51].

Under glare conditions, CT values were higher (worse) than without glare at medium and high frequencies, as previously described in healthy subjects of similar ages [47]. In our study, the effect of lutein supplementation is mainly observed at the medium and high frequencies in the older group, but it did not reach statistical significance, probably because of the large variability of CT values [41,47,49,50,51] (higher in the older group, as previously described [47]) and because of the duration of the supplementation period (two months), which may have been too short to obtain variations in healthy subjects [41,49,50], although a ceiling in serum lutein concentrations was reached before the first month of supplementation. The sample size may also have been relatively small, although a similar sample size was used in another study to obtain variations in contrast sensitivity [15].

On the other hand, although there were significant correlations between serum lutein markers (lutein, lutein plus zeaxanthin) and CT, with and without glare, at the three frequency levels in the total group, both at baseline and after supplementation, these correlations differed when comparing age groups. At baseline, inverse correlations between CT and lutein plus zeaxanthin or lutein/cholesterol (high frequency) and between CT and lutein/cholesterol (low frequency) were found only in the older group and with glare, implying an improvement in contrast sensitivity associated with higher serum lutein and zeaxanthin concentrations. This is in line with previous data in subjects with similar characteristics, where inverse correlations were also obtained at the three frequency levels, but without glare conditions [47]. This desirable association between CT and serum lutein plus zeaxanthin at baseline in the older group did not occur after the lutein supplementation period, as no association was found in the older group. In contrast, high-contrast sensitivity has been described in older women taking lutein plus zeaxanthin supplements (at various doses from 1 mg/d) for over fifteen years in eyes without pathology [48]. However, in the younger group, where no associations were found at baseline, there were several significant associations after lutein supplementation, with and without glare, with lutein and lutein/cholesterol at low, medium, and high frequencies. We do not have an explanation for these different responses to lutein supplementation in younger and older subjects, but circulating lipids (although cholesterolaemia was within the normal range in this study), LDL, different proportions of circulating HDL and LDL cholesterol, possible oxidative modifications of LDL and HDL cholesterol [52], and variations in the lipoprotein receptors that may be selective for different ratios of LDL and HDL [53] and may have an effect on retinal pigment epithelial cells cannot be excluded.

## 5. Conclusions

We can conclude that a daily intake of 6 mg of lutein, from marigold lutein extracts (free and ester forms), for 15 days significantly increased serum lutein concentrations in normolipemic adults to levels associated with a lower risk of age-related eye disease, regardless of the chemical form of lutein supplied. Zeaxanthin, which is also present in marigold extract but at much lower concentrations, also produced a significant increase in its serum concentration. Longer supplementation, up to two months, does not significantly modify the serum levels achieved, as serum lutein reaches a plateau concentration but may contribute to an increase in MPOD (long-term marker of lutein status) and thus improve the effect on the visual outcomes, especially in older subjects, in whom inverse strong associations between serum lutein and CT, mainly in glare conditions, were observed in this study.

## Figures and Tables

**Figure 1 nutrients-16-01415-f001:**
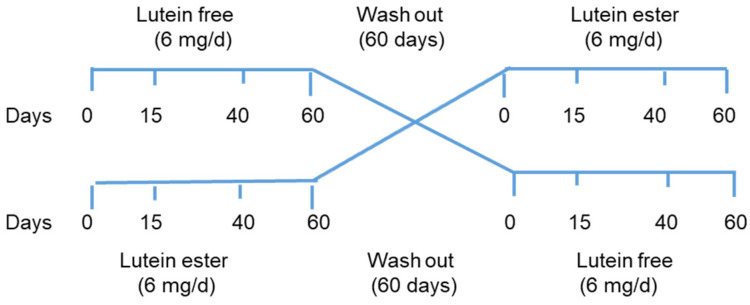
Study design *n* = 24, divided into two groups: lutein-free, *n* = 12 (*n* = 6, three women and three men aged 20–35 y, and *n* = 6, three women and three men aged 50–65 y) and lutein ester, *n* = 12, with the same distribution as in the lutein free group). Blood samples were taken at each time point, CT was assessed at 0 and 60 days, and food records were taken at the start of each period.

**Figure 2 nutrients-16-01415-f002:**
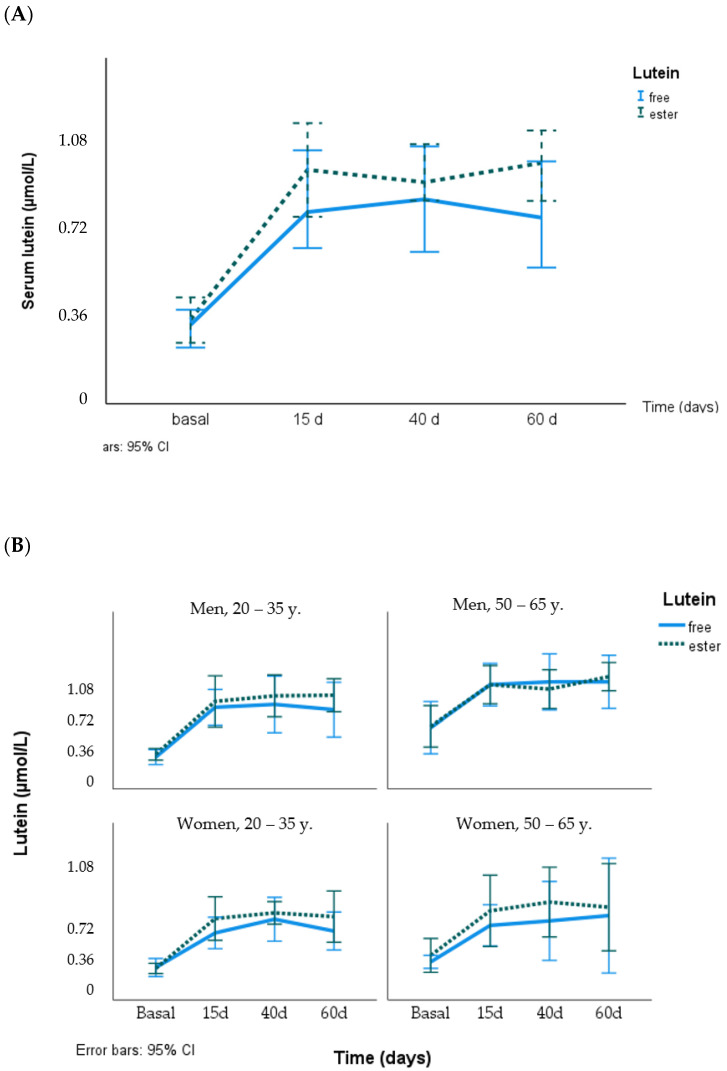
Response of serum lutein concentration (µmol/L) to supplementation with free lutein and lutein esters in the total group (**A**) and grouped by age and sex (**B**).

**Figure 3 nutrients-16-01415-f003:**
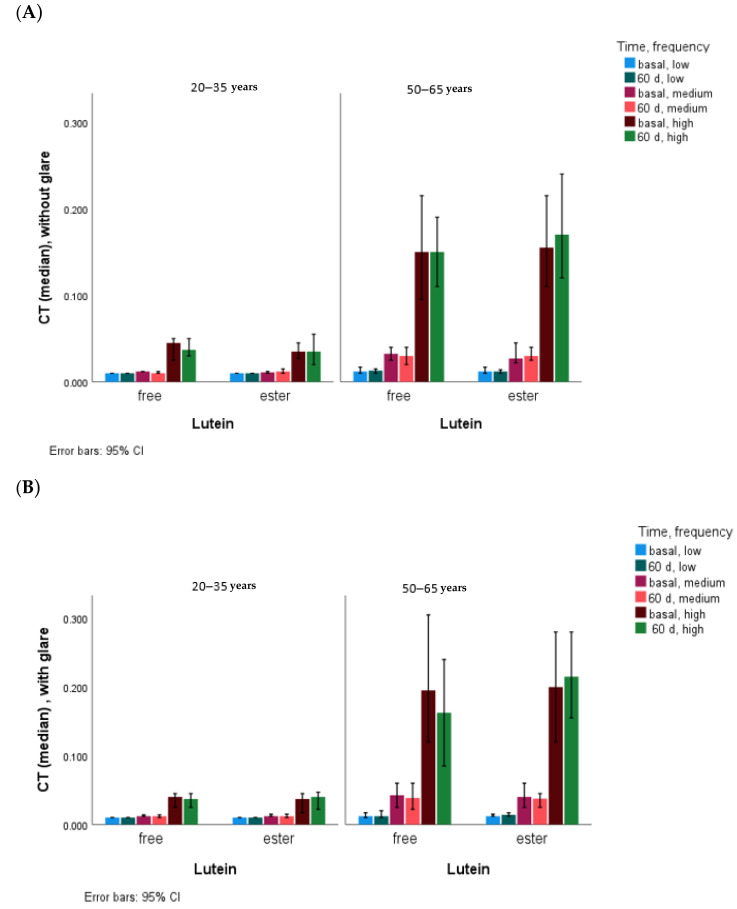
CT values without glare (**A**) and with glare (**B**) according to age and chemical form of lutein. Statistical differences (*p* < 0.001) at all frequency levels, with and without glare, at baseline and at 60 days between younger and older subjects.

**Table 1 nutrients-16-01415-t001:** Serum lutein, zeaxanthin (µmol/L), and lutein/cholesterol concentrations in participants, expressed as mean (SE) [95%CI] throughout the study.

	Free Lutein Group (*n* = 24)			Lutein Ester Group (*n* = 24)			Total Group (*n* = 48)		
	Lutein	Lutein/Cholesterol	Zeaxanthin	Lutein	Lutein/Cholesterol	Zeaxanthin	Lutein	Zeaxanthin	Lutein + Zeaxanthin
Basal	0.35 ^a^ (0.03) [0.29, 0.42]	0.11 ^a^ (0.01) (0.09, 0.13]	0.09 ^a^ (0.01) [0.07, 0.11]	0.38 ^a^ (0.03) [0.32, 0.44]	0.12 ^a^ (0.47) [0.10, 0.14]	0.10 ^a^ (0.01) [0.08, 0.12]	0.37 ^a^ (0.03) [0.31, 0.42]	0.10 ^a^ (0.01) [0.08, 0.11]	0.51 ^a^ (0.04) [0.43, 0.59]
15 d	0.81 ^b^ (0.04) [0.72, 0.89]	0.26 ^b^ (0.02) [0.22, 0.29]	0.14 ^b^ (0.01) [0.12, 0.16]	0.90 ^b^ (0.06) [0.79, 1.01]	0.29 ^b^ (0.02) [0.25, 0.33]	0.14 ^b^ (0.01) [0.13, 0.16]	0.85 ^b^ (0.04) [0.78, 0.93]	0.14 ^b^ (0.01) [0.13, 0.15]	1.02 ^b^ (0.05) [0.92, 1.12]
40 d	0.87 ^b^ (0.06) [0.74, 1.00]	0.29 ^b^ (0.02) [0.24, 0.34]	0.16 ^bc^ (0.01) [0.13, 0.18]	0.94 ^b^ (0.05) [0.84, 1.04]	0.31 ^b^ (0.02) [0.27, 0.36]	0.16 ^bc^ (0.01) [0.14, 0.18]	0.91 ^b^ (0.04) [0.83, 0.99]	0.16 ^c^ (0.01) [0.14, 0.17]	1.10 ^c^ (0.05) [1.03, 1.20]
60 d	0.84 ^b^ (0.07) [0.69, 0.99]	0.27 ^b^ (0.03) [0.22, 0.33]	0.16 ^b^ (0.01) [0.13, 0.18]	0.96 ^b^ (0.06) [0.84, 1.07]	0.31 ^b^ (0.02) [0.27, 0.04]	0.17 ^b^ (0.01) [0.15, 0.19]	0.90 ^b^ (0.05) [0.80, 1.00]	0.16 ^c^ (0.01) [0.15, 0.18]	1.08 ^c^ (0.06) [0.96, 1.20]

^a,b,c^ In the columns, different superscripts indicate statistical significance (*p* < 0.05).

**Table 2 nutrients-16-01415-t002:** Contrast thresholds (mean ± SD [median]) at different degrees of visual angle, with and without glare (*n* = 96 eyes), at baseline and at the end of the study.

	Contrast Threshold		
Visual Angle of the Estimulus (°)	without Glare	with Glare	
	Baseline	Baseline	*p*-value
6.3° and 4.0° (mean)	0.013 ± 0.013 [0.010]	0.013 ± 0.008 [0.010] ^a^	0.145
2.5° and 1.6 (mean)	0.026 ± 0.038 [0.029]	0.037 ± 0.056 [0.017]	<0.001
1.0° and 0.7° (mean)	0.108 ± 0.108 [0.050]	0.118 ± 0.109 [0.060]	0.009
	60 days	60 days	
6.3° and 4.0° (mean)	0.016 ± 0.038 [0.010]	0.014 ± 0.011 [0.010] ^b^	0.050
2.5° and 1.6 (mean)	0.027 ± 0.044 [0.016]	0.033 ± 0.046 [0.020]	<0.001
1.0° and 0.7° (mean)	0.105 ± 0.010 [0.055]	0.112 ± 0.105 [0.070]	0.040

Within columns, different superscripts indicate significant difference between baseline and 60 days (^ab^ *p* = 0.008).

**Table 3 nutrients-16-01415-t003:** Correlations (Spearman’s rho and [*p*-value]) of contrast thresholds (mean at each of the three frequency levels, with and without glare) with serum lutein, lutein plus zeaxanthin, and lutein/cholesterol in the total group. Statistically significant results are shown in italics.

Visual Angle of the Estimulus (°)	Lutein	Lutein + Zeaxanthin	Lutein/Cholesterol	Lutein	Lutein + Zeaxanthin	Lutein/Cholesterol
	Baseline—without glare			Baseline—with glare		
6.3° and 4.0°	*0.345 [0.001]*	*0.329 [0.002]*	0.059 [0.591]	*0.263* *[0.010]*	*0.220 [0.031]*	−0.001 [0.989]
2.5° and 1.6	*0.376* *[<0.001]*	*0.293 [0.007]*	0.063 [0.569]	*0.428* *[<0.001]*	*0.376 [<0.001]*	0.120 [0.245]
1.0° and 0.7°	*0.364 [<0.001]*	*0.264 [0.015]*	0.086 [0.434]	*0.359 [<0.001]*	*0.325 [0.001]*	0.110 [0.284]
60 days—without glare				60 days–with glare		
6.3° and 4.0°	*0.239 [0.026]*	*0.231 [0.032]*	0.019 [0.862]	0.188 [0.067]	*0.210 [0.040]*	−0.031 [0.762]
2.5° and 1.6	*0.262 [0.015]*	*0.276 [0.010]*	−0.039 [0.720]	*0.213 [0.037]*	*0.254 [0.013]*	−0.061 [0.554]
1.0° and 0.7°	0.170 [0.117]	*0.232 [0.032]*	−0.095 [0.386]	*0.322* *[0.001]*	*0.340 [<0.001]*	0.062 [0.547]

## Data Availability

The original contributions presented in the study are included in the article/Appendix A, further inquiries can be directed to the corresponding author.

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
