# Peer review of "Bioavailability of Lutein from Marigold Flowers (Free vs. Ester Forms): A Randomised Cross-Over Study to Assess Serum Response and Visual Contrast Threshold in Adults"

_nutrients, 2024, doi:10.3390/nu16101415_

Round 1

Reviewer 1 Report

Comments and Suggestions for Authors

In this manuscript, the study investigated the effects of lutein intake on the risk of eye disease in young and elderly volunteers, and found that continuous intake of lutein for 15 days increased serum lutein content, thereby reducing the risk of age-related eye disease. There are 24 volunteers in total involved in this study, and the data have reference value and guiding significance for people's lutein intake. However, there exist a lot of studies on the effects of lutein on vision, and there are already a large number of articles that are highly similar to this paper. Comparably, the experimental design in this paper may lack some innovation, and the data in this paper are also not significant enough, which may due to the insufficient number of volunteers.

Places requiring revision are given below:

Line21:(12 women, 12 men), same as line 85

Line153 and Line154: 10 cd/m2, do significant digits need to be uniform?

Line194: (p<0.001), Italics need to be unified

Line215-line218: The two diagrams need to be perfected (font, horizontal and vertical headings), and the error line span is large

Line231-line237: Two images need to be optimized (font, image size, uniformity, gridlines)

Line252: The p in the Table 3 is sometimes absent

Line399: Please delete a space

Comments on the Quality of English Language

The grammar structure is smooth, the expression is clear, and the technical terms and scientific expressions are used. But somewhat wordy, and the logic of the language could be clearer.

Author Response

REVIEWER 1

           In this manuscript, the study investigated the effects of lutein intake on the risk of eye disease in young and elderly volunteers, and found that continuous intake of lutein for 15 days increased serum lutein content, thereby reducing the risk of age-related eye disease. There are 24 volunteers in total involved in this study, and the data have reference value and guiding significance for people's lutein intake. However, there exist a lot of studies on the effects of lutein on vision, and there are already a large number of articles that are highly similar to this paper. Comparably, the experimental design in this paper may lack some innovation, and the data in this paper are also not significant enough, which may due to the insufficient number of volunteers.

Thank you for your review and your helpful comments.

I agree with the reviewer that there are numerous studies on the effects of lutein on vision, but there are very few that consider the chemical form of the lutein supplementation (free vs ester) and their results are controversial.  Regarding the number of volunteers (n=24 in the present study), other studies have estimated and recruited fewer volunteers to assess differences in response to two lutein formulations (e.g. eighteen subjects, Bowen et al., 2002; twenty subjects, Yoshizako et al., 2016), but as the reviewer suggests, it would be interesting to study in a large sample.

Places requiring revision are given below:

 - Line 21(12 women, 12 men), same as line 85

“12 men” was added in line 21, as requested.

- Line 153 and Line 154: 10 cd/m2, do significant digits need to be uniform?

A typo has been detected and now reads: “…. the stimuli were presented at 10 cd/m2. Stimulus presentation lasted 0.2 seconds with 0.2 second intervals; test luminance with a glare of 40,000 cd/m2.”

 - Line 194: (p<0.001), Italics need to be unified. 

Done. p is now in italics throughout the manuscript.  In Table 3, the p value is in italics when there is statistical significance.

- Line 215-line 218: The two diagrams need to be perfected (font, horizontal and vertical headings), and the error line span is large.

Figure 2 has been revised and made uniform (font, horizontal and vertical headings). Error lines are correct.

- Line 231-line 237: Two images need to be optimized (font, image size, uniformity, gridlines)

The images in Figure 3 has been optimised (font, image size) and the grid lines have been removed.

- Line 252: The p in the Table 3 is sometimes absent

The p-value is now in brackets and the title has been changed to make it clearer.

- Line 399: Please delete a space.  Done.

- Comments on the Quality of English Language

the grammar structure is smooth, the expression is clear, and the technical terms and scientific expressions are used. But somewhat wordy, and the logic of the language could be clearer. 

We have shortened some sentences in the discussion to make the text clearer.

Reviewer 2 Report

Comments and Suggestions for Authors

This manuscript presents a bioavailability study of lutein (free and esterified with fat acids) after two months of intake of marigold (Tagetes erecta) flowers by 24 healthy subjects, with a 6 mg diet of lutein (3 times their mean dietary intake). Serum samples were analysed by HPLC.

Some parts of the study are very confusing, such as the use of marigold.

Table 1 is very confusing, out of the proper position.

Some graphics, such as 2.a, bring text in Spanish. They do not allow a proper analysis of the significance of the data, with huge error bars.

Figure 3 needs a better resolution and significant explanation. Did the authors observe any effects?

The discussion is the better-formulated part of the manuscript, more clear, but somewhat repetitive and not focused on the discussion of the data obtained.

CT was not analysed since it is out of my expertise. Indeed, it appears to present the most relevant results.

References are out of format and only a few were published in the last 5 years (2019-23): 12/53; with about 10 references (10/53, almost 20%) of selfcitations.

Comments on the Quality of English Language

None.

Author Response

REVIEWER 2

Thank you for your review and your helpful comments.

- Some parts of the study are very confusing, such as the use of marigold.

The following sentence has been added to lines 110-113 to clarify the use of Flora GLO and Xangold marigold: “The overall composition of the free lutein capsules was: a) Flora GLO 10 % VG 22.22%, cellulose microcrystalline 77.04% and magnesium stearate 2% and that of the lutein ester capsules was: Xangold 10% 43.33%, cellulose microcrystalline 55.93% and magnesium stearate 0.74%”.

- Table 1 is very confusing, out of the proper position.

Table 1 is now in the correct position.

- Some graphics, such as 2.a, bring text in Spanish. They do not allow a proper analysis of the significance of the data, with huge error bars.

The Spanish text was translated into English.

More information (standard error and 95% confidence interval) is given in Table 1.

- Figure 3 needs a better resolution and significant explanation. Did the authors observe any effects?

Figure 3 has now been expanded to allow better visualisation of the results.

The following has been added to the title of Figure 3: After 60 d of lutein supplementation, CT (age adjusted) differed only in the glare condition for the low frequencies (p=0.008)”.

CT showed no significant differences at baseline and in response to lutein supplementation (lines 227-229).

The sentence (now in lines 238 – 239) was changed to read: “Although CT differed by age at medium and high frequencies, there were no differences after adjustment for age (20-25 vs 50-65 years)”.   

- The discussion is the better-formulated part of the manuscript, more clear, but somewhat repetitive and not focused on the discussion of the data obtained.

We have shortened some sentences in the discussion to make the text clearer.

- References are out of format and only a few were published in the last 5 years (2019-23): 12/53; with about 10 references (10/53, almost 20%) of selfcitations.

The format of the references was taken from the Nutrients template, where the margins in this section are narrower than in the rest of the text.

Four of the twelve references published in the last five years correspond to review articles published in the last three years. These articles include relevant publications up to the year 2022.

The self-citacions are because I have been involved in three of the review articles in recent years and also because I have been working on carotenoids for more than thirty years and it is difficult not to refer to previous work.

Reviewer 3 Report

Comments and Suggestions for Authors

This study investigated the relationship between blood lutein concentration and visual contrast threshold through an intervention trial using lutein supplements (random crossover trial, free form vs. ester form).

There are some points that are unclear regarding the experimental method, so the reviewer points them out.

 1. The blood sample appears to be “serum,” but there appears to be no description of how to prepare serum in either this manuscript or Reference 32.

 2. HPLC analysis conditions

Was it possible to separate lutein and zeaxanthin using the ODS column under the HPLC conditions used in this study?

 3. There are some concerns regarding lutein supplement capsules as described below.

 i) Line 354, “In the present study, both lutein supplements had the same formulation (crystalline in oil).”

It is true that in Flora GLO 10%, lutein is in free form and crystalline, but in Xangold® 10%, lutein is in ester form and appears to be beadlets (water dispersible beadlets).

When comparing crystals and beadlets, beadlets appear to have overwhelmingly higher bioaccessibility.

 ii) Did the authors analyze the lutein in supplements?  

Or is it being analyzed by the company the authors commissioned to manufacture the capsules?

There is a possibility that the concentration of ingredients announced by the manufacturer is the analysis result at the time of manufacture at some point, and after that, even if the material or lot is different, no new analysis is performed and the analysis value remains as it is. Isn't there a lot of things like that? In that case, there remains some concern as to whether the lutein concentration in the capsules actually used in the experiment was really 6 mg.

 iii) It is said that the cis form of carotenoids has higher bioaccessibility than the trans form. Is the ratio of cis and trans forms the same in supplements? In this respect as well, it seems necessary to analyze lutein on the supplement.

 iv) If the lutein content of both supplements is 6mg,

The number of moles in 6mg is the molecular weight of free lutein is 568.8714, so it is 0.011 mmol (11 umol),the molecular weight of dipalmitin ester of lutein is 1045.7314, so it is 0.0057 mmol (5.7 umol).

 If both the free form and the ester form of lutein are 6 mg, the ester form has about half the mole number of the free form. In this case, the number of moles would be 2 vs. 1, meaning the free form would be dosed twice as much as the ester form?  

 Literature (J Chromatography, 262, 1983, 277-284)

The fatty acids ester-bonded to marigold lutein are myristicin, palmitin, and stearin.

v) Also, if the two types are compared, it seems that the conditions would not be the same unless it was lutein free form + fatty acid free form x 2 molecules vs fatty acid diester form of lutein.

Author Response

REVIEWER 3

There are some points that are unclear regarding the experimental method, so the reviewer points them out.

Thank you for your review and your helpful comments.

  1. The blood sample appears to be “serum,” but there appears to be no description of how to prepare serum in either this manuscript or Reference 32.

The following sentence has been added on lines 138-139: “Blood samples were collected in plain blood collection tubes without anticoagulant. The serum was separated by centrifugation (630 x g, 10 min)”.

  1. HPLC analysis conditions

Was it possible to separate lutein and zeaxanthin using the ODS column under the HPLC conditions used in this study?

Separation of lutein and zeaxanthin is possible using the column and conditions described in this manuscript, although resolution is better with the C30 column. We have published several papers on lutein and zeaxanthin using this chromatographic method, e.g.:

Olmedilla et al. Brit J Nutr., 2001; 85:227-238 https://pubmed.ncbi.nlm.nih.gov/11242491/ ;

Olmedilla et al. Clin Sci 2002; 447-456    https://pubmed.ncbi.nlm.nih.gov/11914107/ ;

Olmedilla-Alonso et al., 2005; 38:444-459  https://www.sciencedirect.com/science/article/pii/S0009912005000433 ;

Olmedilla-Alonso et al. Nutr J. 2014, 13:52, http://www.nutritionj.com/content/13/1/52

  1. There are some concerns regarding lutein supplement capsules as described below.
  2. i) Line 354, “In the present study, both lutein supplements had the same formulation (crystalline in oil).”

It is true that in Flora GLO 10%, lutein is in free form and crystalline, but in Xangold® 10%, lutein is in ester form and appears to be beadlets (water dispersible beadlets).

When comparing crystals and beadlets, beadlets appear to have overwhelmingly higher bioaccessibility.

Thank you for your comment. Xangold 10% was not as beadlets (now corrected in the text). I checked the specifications sent by Cognis and the product description was: “Xangold® 10% is a dark, reddish-brown microencapsulated tablet grade powder containing natural mixed carotenoid esters isolated from marigold flowers (Tagetes erecta)” and “Solubility: insoluble in water, slightly soluble in organic solvents”. Thus, the sentence in line 354 has been deleted.

  1. ii) Did the authors analyze the lutein in supplements?  

Or is it being analyzed by the company the authors commissioned to manufacture the capsules?

There is a possibility that the concentration of ingredients announced by the manufacturer is the analysis result at the time of manufacture at some point, and after that, even if the material or lot is different, no new analysis is performed and the analysis value remains as it is. Isn't there a lot of things like that? In that case, there remains some concern as to whether the lutein concentration in the capsules actually used in the experiment was really 6 mg.

No, we have not analysed the lutein in the capsules and, as in most published studies, we have used the composition reported by the supplier and manufacturer. However, you are right, the concentration can vary with storage time, but I have not had the opportunity to analyse the capsules at that time. We analysed lutein capsules (from marigold extract) in another study and the composition (from saponified extract) was: trans lutein 12.1 mg/capsule and 13/15-cis-lutein 3.2 mg/capsule (reference 44: Olmedilla et al. A European multicenter, placebo-controlled supplementation study with a-tocopherol, carotene-rich palm oil, lutein or lycopene: analysis of serum responses. Clin Sci 1002; 102: 447-456.)

 The following composition of the capsules (according Goerlich) has been added to the lines 110-113: “The overall composition of the free lutein capsules was: a) Flora GLO 10 % VG 22.22%, cellulose microcrystalline 77.04% and magnesium stearate 2% and that of the lutein ester capsules was: Xangold 10% 43.33%, cellulose microcrystalline 55.93% and magnesium stearate 0.74%”.

 iii) It is said that the cis form of carotenoids has higher bioaccessibility than the trans form. Is the ratio of cis and trans forms the same in supplements? In this respect as well, it seems necessary to analyze lutein on the supplement.

Marigold flowers (Tagetes erecta) are rich in lutein, mainly in the form of lutein esters (90-99%)[5] (lines 65-67). According to Li et al, the lutein esters in marigold flowers (from 11 cultivars of Tagetes erecta L) were predominantly composed of six all trans-diesters, but small amounts of cis isomers of the respective diesters were also present (Li W, Gao Y, Zhao J, Wang Q. Phenolic, flavonoid, and lutein ester content and antioxidant activity of 11 cultivars of chinese marigold. J Agric Food Chem. 2007 Oct 17;55(21):8478-84). On the other hand, it is generally accepted that most carotenoids occur in nature as all-trans isomers (Nagy, V.; Agócs, A.; Balázs, V.L.; Purger, D.; Filep, R.; Sándor, V.; Turcsi, E.; Gulyás-Fekete, G.; Deli, J. Lutein Isomers: Preparation, Separation, Structure Elucidation, and Occurrence in 20 Medicinal Plants. Molecules 2023, 28, 1187).  However, I agree with you that isomerisation of all-trans-lutein to cis forms occurs due to environmental conditions and then, storage temperature, light, etc. could facilitate the degradation of all-trans-lutein, but the capsules were manufactured, delivered and stored in our laboratory under appropriate conditions and for a very short period of time.

  1. iv) If the lutein content of both supplements is 6 mg,

The number of moles in 6mg is the molecular weight of free lutein is 568.8714, so it is 0.011 mmol (11 umol), the molecular weight of dipalmitin ester of lutein is 1045.7314, so it is 0.0057 mmol (5.7 umol).

 If both the free form and the ester form of lutein are 6 mg, the ester form has about half the mole number of the free form. In this case, the number of moles would be 2 vs. 1, meaning the free form would be dosed twice as much as the ester form?  

 Literature (J Chromatography, 262, 1983, 277-284)

The fatty acids ester-bonded to marigold lutein are myristicin, palmitin, and stearin.

We agree with your comment. The following composition of the capsules has been added to the lines 110-113: “The overall composition of the free lutein capsules was: a) Flora GLO 10 % VG 22.22%, cellulose microcrystalline 77.04% and magnesium stearate 2% and that of the lutein ester capsules was: Xangold 10% 43.33%, cellulose microcrystalline 55.93% and magnesium stearate 0.74%”.

  1. v) Also, if the two types are compared, it seems that the conditions would not be the same unless it was lutein free form + fatty acid free form x 2 molecules vs fatty acid diester form of lutein.

Although we believe that the two forms of lutein were dosed correctly (see previous answers). However, although the source of lutein was the same, marigold flowers, the suppliers are different and there could be factors that influence its bioavailability, as well as factors related to the overall diet in which the subjects incorporated the intake of lutein capsules.

Round 2

Reviewer 1 Report

Comments and Suggestions for Authors

I have carefully read this research paper on the bioavailability of lutein in calendula. The study compared the effects of free and esterified forms of lutein on serum response and visual contrast thresholds in adults through a randomized cross-over design. Overall, the study's rigorous design, sound methodology, and detailed data analysis provide valuable insights into the bioavailability of lutein and its potential benefits for visual function. However, before considering publication, I suggest that the authors revise and add the following:

The introduction section of the article mentions the deposition of lutein and zeaxanthin in the blood and retina and their association with chronic disease risk. It is recommended that the authors further discuss the mechanisms by which these compounds are metabolized and transported in the body, and why they are particularly important for eye health.

The study method mentioned the use of 7-day food records to determine dietary intake. Authors are advised to detail how food records were ensured to be accurate and representative, and what impact this might have had on study results.

Methods for determining serum lutein and zeaxanthin concentrations are mentioned, but possible analytical errors or variations are not mentioned. It is recommended that authors discuss possible sources of error during the experiment and show how these errors can be controlled through experimental design and statistical analysis.

The results showed in part that the effects of lutein supplementation on the visual contrast threshold also did not differ at any point in time. However, there may be differences between these findings and those of previous studies. The authors are advised to compare and explain these differences in the discussion section, as well as the implications this may have for clinical practice and future research directions.

The conclusions of the paper highlight the potential benefits of lutein supplementation in reducing the risk of age-related eye disease. It is recommended that the authors provide more discussion on how to translate these findings into public health recommendations and interventions in the conclusion section.

The charts and data in the article are presented clearly, but authors are advised to check the titles and legends of the charts to make sure they are accurate and easy to understand.

Please proofread the full text carefully to correct any grammatical errors and typos to ensure professionalism and readability.

Overall, this study provides important information for our understanding of the bioavailability of lutein and its potential benefits for visual health. After considering the above recommendations, I believe this article will be of benefit to both researchers and clinicians in the field.

Author Response

Comments and Suggestions for Authors

I have carefully read this research paper on the bioavailability of lutein in calendula. The study compared the effects of free and esterified forms of lutein on serum response and visual contrast thresholds in adults through a randomized cross-over design. Overall, the study's rigorous design, sound methodology, and detailed data analysis provide valuable insights into the bioavailability of lutein and its potential benefits for visual function. However, before considering publication, I suggest that the authors revise and add the following:

1) The introduction section of the article mentions the deposition of lutein and zeaxanthin in the blood and retina and their association with chronic disease risk. It is recommended that the authors further discuss the mechanisms by which these compounds are metabolized and transported in the body, and why they are particularly important for eye health.

REPLY –  The following paragraph is added to line 43: “Lutein and its structural isomer zeaxanthin are obtained exclusively from food. These carotenoids can be absorbed intact or undergo oxidative cleavage prior to absorption from the intestinal lumen and are transported in the blood by lipoproteins, associated with LDL and HDL lipoproteins. Lutein and zeaxanthin are preferentially deposited in the retina where, together with meso-zeaxanthin (thought to be formed from lutein in the retina), they form the macular pigment (MP). This specific intraocular deposition suggests a biological process governing their capture, deposition and stabilisation in the macula, thought to be mediated by binding proteins”.

The following sentence is added to line 65: “In the eye, MP filters (or absorbs) blue light, physically protecting the underlying photoreceptor cell layer from light damage, but also acts as an antioxidant to protect against the formation of reactive oxygen species [1,15]”.

2) The study method mentioned the use of 7-day food records to determine dietary intake. Authors are advised to detail how food records were ensured to be accurate and representative, and what impact this might have had on study results.

REPLY- The following sentence has been added to line 118: “Food records were kept for seven consecutive days (including one weekend) and were collected by a specialised dietician who reviewed the records in the presence of the participant, asking questions about quantities or items to facilitate correct identification of the food /drink consumed”.  

3) Methods for determining serum lutein and zeaxanthin concentrations are mentioned, but possible analytical errors or variations are not mentioned. It is recommended that authors discuss possible sources of error during the experiment and show how these errors can be controlled through experimental design and statistical analysis.

REPLY –  The following paragraph was added to line 158: “Baseline and supplemented serum samples from each subject were stored at 70 ºC, processed and analysed simultaneously to reduce analytical variability. Subjects were analysed in random order within seven months of collection and one in six serum samples was analysed in duplicate”.

 4) The results showed in part that the effects of lutein supplementation on the visual contrast threshold also did not differ at any point in time. However, there may be differences between these findings and those of previous studies. The authors are advised to compare and explain these differences in the discussion section, as well as the implications this may have for clinical practice and future research directions.

REPLY -  The paragraphs in lines 411-424 have been changed and now read:

“CT was influenced by lutein supplementation, mainly under glare conditions and at low frequencies, and as in previous studies, this relationship appears to be age-dependent [47], found only in the older group, where an increase in the CT (lower contrast sensitivity) was obtained after two months of lutein supplementation. However, in the study by Yao et al. [49], a slight trend towards an improvement in the glare sensitivity (mainly at medium frequencies) was observed from the first to the third month of supplementation (20 mg/day), and only after six months of lutein supplementation was a significant change observed in healthy subjects. Also in young subjects without chronic eye disease, Ma et al. [50] reported a trend toward higher thresholds at low and medium frequencies after three months of supplementation with 6 mg lutein /day, but without improvement in glare conditions. Another study in healthy subjects (20-69 years) supplemented with 12 mg lutein /day for four months showed an improvement in contrast sensitivity (low frequency) and glare sensitivity (high and medium frequency) [41]. In subjects with age-related macular degeneration, lutein supplementation at different doses (6, 10 and 20 mg/d) with visual function assessments at six, nine and twelve months showed no significant changes until the twelve months [15,51].

Under glare conditions, CT values were higher (worse) than without glare at medium and high frequencies, as previously described in healthy subjects of similar age [47]. In our study, the effect of lutein supplementation is mainly observed at the medium and high frequencies in the older group, but it did not reach statistical significance, probably because of the large variability of CT values [41,47,49,50,51] (higher in the older group, as previously described [47-Estévez]) and because of the duration of supplementation period (two months) may have been too short to obtain variations in healthy subjects [41,49,50], although a ceiling in serum lutein concentrations was reached before the first month of supplementation. The sample size may also have been relatively small, although a similar sample size was used in another study to obtain variations in contrast sensitivity [15]. “

  1. Yao, Y.; Qiu, A-h.; Wu, S-W-; Cai, Z-y.; Xu, S.; Liang, X-q. Lutein supplementation improves visual performance in Chinese drivers: 1-year randomized, double-blind, placebo-controlled study. Nutrition, 2013, 29, 958-964.
  2. Ma, L.; Lin, X-M.; Zou, A-Y.; Xu, X-R.; Li, Y.; Xu, R. A 12-week lutein supplementation improves visual function in Chinese people with long-term computr display light exposure. Brit. J. Nutr., 2009, 102, 186-190.
  3. Ma, L.; Yan, S-F.; Huang, Y-H.; Lu, X-R.; Qian, F.; Pan, H-L.; et al. Effect of lutein and zeaxanthin on mcular pigment and visual function in patients with early age-related macular degeneration. Ophthalmol. 2012, 119, 2290-2297.

5) The conclusions of the paper highlight the potential benefits of lutein supplementation in reducing the risk of age-related eye disease. It is recommended that the authors provide more discussion on how to translate these findings into public health recommendations and interventions in the conclusion section.

REPLY – In the conclusions, lines 470-472: “…. Daily intake of 6 mg of lutein …. For 15 days significantly increased serum lutein concentrations …… to levels associated with a lower risk of age-related eye disease, …..”. That is, the levels of serum lutein achieved in this study are similar to those associated with a lower risk of age-related eye disease in the literature, but it was not our aim to highlight the potential benefits of lutein supplementation in reducing this risk.

In terms of public health recommendations, we feel that this is outside the aim of our study, but if you feel it is necessary, we could try to make some suggestions.

6) The charts and data in the article are presented clearly, but authors are advised to check the titles and legends of the charts to make sure they are accurate and easy to understand.

REPLY – The titles and legends of figures and tables have been checked and changes have been made to the titles of Figures 1 and 3 and Table 3.  

 7) Please proofread the full text carefully to correct any grammatical errors and typos to ensure professionalism and readability.

REPLY – Done.

8) Overall, this study provides important information for our understanding of the bioavailability of lutein and its potential benefits for visual health. After considering the above recommendations, I believe this article will be of benefit to both researchers and clinicians in the field.

REPLY – Thank you for your comment.

Reviewer 2 Report

Comments and Suggestions for Authors

The revised version was presented, and it appears that minimal changes have been made since the previous submission. The alterations are primarily confined to minor adjustments in the titles of figures, with limited modifications observed in the text, particularly in the paragraph located at line 296. Despite the specific aspects highlighted for improvement, it seems that these recommendations have not been adequately addressed. As a result, the concerns raised in the previous review remain largely unattended.

Comments on the Quality of English Language

None.

Author Response

Comments and Suggestions for Authors

The revised version was presented, and it appears that minimal changes have been made since the previous submission. The alterations are primarily confined to minor adjustments in the titles of figures, with limited modifications observed in the text, particularly in the paragraph located at line 296. Despite the specific aspects highlighted for improvement, it seems that these recommendations have not been adequately addressed. As a result, the concerns raised in the previous review remain largely unattended.

As reviewer 2 feels that the specific aspects highlighted for improvement have not been adequately addressed and we have not followed her/his recommendations, I will try to expand on the points she/he made in her/his first review.

1)  Some parts of the study are very confusing, such as the use of marigold.

The following sentence was added to lines 110-113 to clarify the use of marigold: “The overall composition of the free lutein capsules was: a) Flora GLO 10 % VG 22.22%, cellulose microcrystalline 77.04% and magnesium stearate 2% and that of the lutein ester capsules was: Xangold 10% 43.33%, cellulose microcrystalline 55.93% and magnesium stearate 0.74%”.

Please, let us know if there are other parts of the study that need to be better explained.

2) Table 1 is very confusing, out of the proper position.

Table 1 is now in the correct position. Please, let us know where it is “very confusing”, we do not know how to make the presentation of serum concentrations at baseline and at each time point clearer.

3)  Some graphics, such as 2.a, bring text in Spanish. They do not allow a proper analysis of the significance of the data, with huge error bars.

The Spanish text was translated into English.

Figure 2a: More information is given in Table 1: standard error, 95% confidence interval and the statistical significance (p<0.05) throughout the 60 d supplementation period.

4)  Figure 3 needs a better resolution and significant explanation. Did the authors observe any effects?

Figure 3 has been enlarged and the y-axes of Figures 3A and 3B now have the same scale (0.0 – 0.3) to allow better visualisation of the results.

The following has been added to the title of Figure 3: “After 60 d of lutein supplementation, CT (age adjusted) differed only in the glare condition for the low frequencies (p=0.008)”.

5) CT showed no significant differences at baseline and in response to lutein supplementation (lines 227-229).

A pair of sentences has been changed in line 229 to read: “In the total group, CT showed no significant differences ……. “ and in lines 234-235:Although CT differed by age at medium and high frequencies, there were no differences after adjustment for age (20-25 vs 50-65 years)”.   

In the title of Figure 3, the following sentence has been added: “After 60 d of lutein supplementation, CT (age adjusted) differed in the glare condition for the low frequencies (p=0.008)”.

6) The discussion is the better-formulated part of the manuscript, more clear, but somewhat repetitive and not focused on the discussion of the data obtained.

REPLY -  The paragraphs in lines 411-424 have been changed and now read:

“CT was influenced by lutein supplementation, mainly under glare conditions and at low frequencies, and as in previous studies, this relationship appears to be age-dependent [47], found only in the older group, where an increase in the CT (lower contrast sensitivity) was obtained after two months of lutein supplementation. However, in the study by Yao et al. [49], a slight trend towards an improvement in the glare sensitivity (mainly at medium frequencies) was observed from the first to the third month of supplementation (20 mg/day), and only after six months of lutein supplementation was a significant change observed in healthy subjects. Also in young subjects without chronic eye disease, Ma et al. [50] reported a trend toward higher thresholds at low and medium frequencies after three months of supplementation with 6 mg lutein /day, but without improvement in glare conditions. Another study in healthy subjects (20-69 years) supplemented with 12 mg lutein /day for four months showed an improvement in contrast sensitivity (low frequency) and glare sensitivity (high and medium frequency) [41]. In subjects with age-related macular degeneration, lutein supplementation at different doses (6, 10 and 20 mg/d) with visual function assessments at six, nine and twelve months showed no significant changes until the twelve months [15,51].

Under glare conditions, CT values were higher (worse) than without glare at medium and high frequencies, as previously described in healthy subjects of similar age [47]. In our study, the effect of lutein supplementation is mainly observed at the medium and high frequencies in the older group, but it did not reach statistical significance, probably because of the large variability of CT values [41,47,49,50,51] (higher in the older group, as previously described [47-Estévez]) and because of the duration of supplementation period (two months) may have been too short to obtain variations in healthy subjects [41,49,50], although a ceiling in serum lutein concentrations was reached before the first month of supplementation. The sample size may also have been relatively small, although a similar sample size was used in another study to obtain variations in contrast sensitivity [15]. “

  1. Yao, Y.; Qiu, A-h.; Wu, S-W-; Cai, Z-y.; Xu, S.; Liang, X-q. Lutein supplementation improves visual performance in Chinese drivers: 1-year randomized, double-blind, placebo-controlled study. Nutrition, 2013, 29, 958-964.
  2. Ma, L.; Lin, X-M.; Zou, A-Y.; Xu, X-R.; Li, Y.; Xu, R. A 12-week lutein supplementation improves visual function in Chinese people with long-term computr display light exposure. Brit. J. Nutr., 2009, 102, 186-190.
  3. Ma, L.; Yan, S-F.; Huang, Y-H.; Lu, X-R.; Qian, F.; Pan, H-L.; et al. Effect of lutein and zeaxanthin on mcular pigment and visual function in patients with early age-related macular degeneration. Ophthalmol. 2012, 119, 2290-2297.

We have shortened some sentences in the discussion to make the text clearer. The text changed in the previous revision is now highlighted or in red (lines 322, 362-364, 395-401).

7) References are out of format and only a few were published in the last 5 years (2019-23): 12/53; with about 10 references (10/53, almost 20%) of selfcitations.

 - The format of the references was taken from the Nutrients template, where the margins in this section are narrower than in the rest of the text.

- There are ten references published in the last five years, six of which are review articles. Although we believe that the information we provide is up to date, if the reviewer feels that we should include an article of particular relevance, we would be grateful if you could let us know.

- Self-citations: We have excluded references 1, 2 and 50.

Other self-citated articles are needed to understand the background and history of the work presented in this manuscript:

                Ref. 6 (now reference 5) corresponds to the data reported in the mentioned Spanish and Brazilian populations.
            Refs. 29 (now reference 27): The references correspond to the original carotenoid content data published and included in the database used to calculate the dietary intake.
            Ref. 38 (now reference 36): review of carotenoid data in studies reporting dietary intake and blood concentrations.
            Ref. 39, 40 (now references 37, 38): references with dietary and blood lutein concentrations in two age groups of Spanish men and women (ref 37) and in a representative Spanish sample (ref 38).
            Ref. 49 (now reference 47): we refer to a previous work in which a large variability of CT values was described.

Reviewer 3 Report

Comments and Suggestions for Authors

Reviewer 3 checked the authors' comments and revised the manuscript.

Reviewer 3 considers the manuscript to be sufficiently revised.

Author Response

Reviewer 3 checked the authors' comments and revised the manuscript.

Reviewer 3 considers the manuscript to be sufficiently revised.

REPLY - Thank you for your review.
